# Premature Mortality Due to Chronic Obstructive Pulmonary Disease (COPD) in Poland

**DOI:** 10.3390/medicina57020126

**Published:** 2021-02-01

**Authors:** Waclaw Moryson, Barbara Stawinska-Witoszynska

**Affiliations:** Department of Epidemiology and Hygiene, Chair of Social Medicine, Poznan University of Medical Sciences, Rokietnicka 4, 60-806 Poznan, Poland; bwitoszynska@ump.edu.pl

**Keywords:** premature mortality, COPD, smoking, epidemiology

## Abstract

*Background and objectives:* Chronic obstructive pulmonary disease (COPD) is responsible for 5.3% of deaths worldwide and constitutes the third most common cause of death. The deaths from this cause occur over 10times more often in smokers than in non-smokers. Fortunately, for nearly 30 years, the proportion of people smoking tobacco in Poland has been decreasing. This study aims to analyse the change in premature mortality rates of men and women due to COPD in Poland during 2008–2017. *Materials and Methods:* The time trends of deaths occurring under 65 were analysed. Standardised premature mortality rates were used, as well as the mortality rates for the five-year age ranges, with a breakdown by gender. The Joinpoint model was used to determine time trends. *Results:* Over the period analysed, premature mortality in the female population decreased by 2.6% from year to year, albeit without statistical significance, and in the male population there was a decrease by statistically significant 5.2% per year. The biggest drop in mortality, almost 10% per year, was observed in the group of females aged between 50 and 54. Among males, the most significant reduction in mortality was observed in groups between the ages of 40 and 54, and it amounted to approximately 8% annually. With increasing age, the dynamics of mortality reduction decreased. *Conclusions:* The study showed a steady downward trend in premature mortality due to chronic obstructive pulmonary disease in Poland in both genders. The reduction in mortality was at a high level, despite the lower mortality due to this cause than in other European countries.

## 1. Introduction

Chronic obstructive pulmonary disease (COPD) is one of the most common causes of morbidity and mortality. In 2016, it contributed to 5.3% of all deaths worldwide, constituting the third highest cause of mortality. In Poland and other highly developed countries, it was responsible for almost the same percentage of deaths (5.4%). However, in the hierarchy of mortality, it was ranked fifth [1].

The emergence and development of COPD among the inhabitants of highly developed countries is mainly a consequence of both active and passive exposure to tobacco smoke [2,3]. It is estimated that smoking reduces life expectancy by 10years [4] and is the most common preventable cause of mortality in the world [5]. On the other hand, in terms of chronic obstructive pulmonary disease, smoking increases the risk of death by more than 10times in the case of females and more than 13times in the case of males, compared to individuals who have never smoked [6].

In recent years, several changes have been observed in Poland, resulting in a reduction in mortality due to non-infectious diseases, i.e., cardiovascular diseases and malignant tumours [7]. In the context of tobacco-dependent diseases, particularly COPD, the implementation of smoke-free policies based on the World Health Organization model (under the acronym MPOWER) has been remarkable [8]. Successive implementation of the solutions based on six main assumptions, i.e., (Monitor) monitoring tobacco consumption and exposure to tobacco smoke, (Protect) protecting from tobacco smoke including a smoking ban in public places, (Offer) offering active anti-smoking guidance, (Warn) warning of the dangers of smoking, (Enforce Smoking ban) banning tobacco advertising, (Raise taxes) raising tobacco taxes, has brought about the reduction in the percentage of regular tobacco smokers in Poland. Thus in the years 2009–2017, a decline in the percentage of smokers from 32% to 29% for males, and from 24% to 20% for females was observed. Also, it has been shown that passive exposure to tobacco smoke has decreased [9].

The actions taken in Poland since the early 1990s leading to a reduction in exposure to many harmful factors, such as high blood cholesterol levels, harmful behaviour concerning alcohol consumption, and nicotinism, are in line with the WHO’s 2013 plan to reduce premature mortality from non-communicable diseases by 25% before 2025 [10,11]. This paper presents an analysis of changes in premature mortality of men and women due to COPD in Poland during 2008–2017.

## 2. Materials and Methods

This paper contains an analysis of all registered deaths of Polish residents under 65 years of age between 2008 and 2017. The data concerning the number of deaths and the size of the population in the age ranges came from the Central Statistical Office. The analysis included deaths for which J40–J44 codes were registered as the initial cause of death, according to the International Statistical Classification of Diseases and Health Problems (ICD-10) [12].

For the analysis of the mortality of male and female populations under 65, the mortality rates standardised based on the 2013 European standard population and crude death rates for age groups, were used [13]. The mortality trends were determined for the five-year age ranges (20–24, 25–29, 30–34, 35–39,40–44, 45–49, 50–54, 55–59, 60–64 years of age), taking into account the division by the gender.

The Joinpoint model was used to determine time trends in mortality. The model constitutes a particular version of linear regression, presenting time trends by means of a segmented curve consisting of the sections that join at points where a change in the time trend is significant. The analysis was conducted with the minimum number of zero joinpoints (straight line), followed by tests for the model fitted with a maximum of one joinpoint. The Monte Carlo Permutation method for tests of significance was used. The homoscedasticity model was selected (errors are assumed to have constant variance). Using the Monte Carlo Permutation technique with 4499 randomly selected data sets, the numbers and locations of the joinpoints, with the best-fitted models for COPD mortality trend, were estimated. Based on the calculated models, annual percentage change (APC) between points and average annual percentage change over the whole analysed period were determined for mortality rates [14]. The Joinpoint results are not shown for the subgroups aged <40 years because at least in one year from the analysed range, the number of deaths due to COPD was zero. A statistical software, Joinpoint Regression 4.7.0.0 (US National Cancer Institute, Rockville, MD, USA), served to determine the regression models. For the annual percentage change (APC) as well as for the average annual percentage change (AAPC), a 95% confidence interval (95% CI) was determined and a value of *p* < 0.05 was assumed as the level of relevance.

## 3. Results

In Poland, between 2008 and 2017, COPD caused 22,841 deaths of females and 46,363 of males. Approximately one-sixth of them were premature, which accounted for 1% of all female and male premature deaths (Table 1).

In the period under consideration, the majority of cases of premature mortality due to COPD affected people over 50, accounting for 97% of premature deaths in women and 95% of premature deaths in men. Moreover, in all age groups, male mortality exceeded female mortality (Table 2).

In 2008, the standardised premature mortality rate due to COPD in Poland was 2.34/100,000 for women and 5.98/100,000 for men. By contrast, standardised mortality rates for the general population were 13.7/100,000 for women and 52.2/100,000 for men. Between 2008 and 2017, there was a steady decline in these numbers (Table 1). In the female population, premature mortality decreased by 2.6% (−5.1%.0.0%; CI) from year to year, albeit without statistical significance, whereas in the male population a statistically significant drop of 5.2% (−6.9%.−3.3%; CI) per year was observed (Figure 1).

The analysis of mortality due to COPD in the abovementioned age groups showed a decrease in mortality rates in all age ranges, except for females between 60 and 64 years of age. Furthermore, the dynamics of mortality reduction showed variability between different age groups.

The most significant decrease in mortality, amounting to almost 10% per year, was observed in the group of women aged between 50 and 54. Among men, the most substantial reduction in mortality was observed in groups between the ages of 40 and 54 and amounted to approximately 8% per year (Table 3). With increasing age, the dynamics of mortality reduction decreased. However, even in the oldest of the groups surveyed, i.e., males between the ages of 60 and 64, the reduction in mortality rate amounted to 5% per year. In the group of females between 60 and 64, the mortality rate increased by 0.6% annually (Table 3). It should be stressed that time trends in mortality over the analysed period were linear in all groups, except for females aged between 40 and 44. However, the considerable variation in the percentage of mortality in individual years in this group, as well as the change in the direction of the trend, must be interpreted with great caution, since deaths due to COPD occur sporadically among women between 40 and 44 (fewerthan 10 cases per year).

## 4. Discussion

In the 1970s and 1980s, Poland was one of the countries with the highest per capita tobacco consumption in the world [15]. At that time, the mortality caused by civilisation diseases, such as cardiovascular diseases or malignant tumours, was at a much higher level in Poland than in other European countries [7]. According to WHO, in 1990, a 15-year-old boy in Poland was less likely to live to the age of 60 than his peer in India or China [16]. The exception in this situation was the COPD mortality, which was lower than the European average [17].

Together with systemic change, the last three decades have brought improvements in premature mortality. In Poland, average life expectancy has increased by 7.9 years for men and 6.6 years for women [18]. The mortality due to ischaemic heart disease and cerebral vascular disease has equalled the European average [19]. Similarly, the mortality associated with most tobacco-dependent cancers has decreased [20]. The study has shown that the level of mortality due to COPD has also been significantly reduced in recent years. However, in contrast to mortality due to cardiovascular diseases and cancer, it was lower at the beginning of the observed period in comparison with other highly developed countries.

Despite the growing knowledge of the development of COPD at a young age [3,21], studies on young people’s mortality are scarce. This study shows an annual decline in mortality before the age of 65 by over 2.5% for females and over 5% for males. The available studies on time trends of COPD mortality in other countries included older patient populations. According to the analysis of COPD mortality in the European population aged 40–85 years, between the years 1994–2010, the EU standardised male mortality rates decreased by 2.6% per year. A reduction in mortality rates of over 5% per year was recorded only in Slovenia (6.6%), France (6.5%), and Ireland (5.3%). The reduction of mortality rates due to COPD in the male population in Poland was, according to this analysis, 0.3% per year. An average annual decline in standarised mortality rates in the female population, between 1994 and 2010, was 0.8% per year. However, the female mortality rates increased in 14 of the 27 analysed countries. A reduction of mortality rates in the female population of over 2.5% per year was recorded in Greece (21.77%), France (7.4%), Lithuania (5.34%), Spain (4.5%), Estonia (4.8%), Malta (3.4%), Romania (3.1%), Slovenia (3.1%), and Bulgaria (2.6%). The standardised mortality rates due to COPD in Poland increased by 1.6% per year [17].

On the other hand, the authors of the COPD mortality analysis in the population between 50 and 84 years of age in the years 1995–2017 indicated a decrease in male mortality in most European countries except for Hungary, where a plateau was observed, as well as the Czech Republic and Croatia, which saw an increase in the COPD-related mortality rate. In the female population, the mortality caused by COPD increased in half of the European countries analysed. In the Czech Republic and Hungary, where the increase was the most rapid, the changes reached 5% per year [2]. It should be noted that the papers assessing mortality rates at the turn of the 1990s and 2000s cover the period when, in most European countries, the International Classification of Disease (ICD) was switched from the ninth to the tenth revision. It is, therefore, possible that the effect of this change was a significant decrease in registered deaths due to COPD. Though such a hypothesis was formulated by the authors of one of the Joinpoint analyses [17], there is no study assessing the impact of the new disease classification on the quality of statistics on deaths due to COPD. Like Poland, Hungary, the Czech Republic, and Croatia are countries of the former Eastern Bloc. Over the past three decades, they have undergone a similar political transformation, resulting in a significant life expectancy improvement. However, based on this study, there appears to be a difference in mortality trends due to COPD among these countries [22].

Raised in several publications, the decreasing excessive mortality of males fromCOPD in many highly developed countries also deserves attention. In some Western countries, this tendency results from a simultaneous reduction in male mortality and an increased female mortality rate [2,17,23]. In the Polish population, the gap in mortality rates between the two sexes was also narrowing, but this was only due to the higher rate of decline in male mortality, compared with female mortality. Nevertheless, in the oldest group of women covered by the above analysis, an increase in mortality rates was observed, which may raise doubts about the direction of changes in mortality in the age groups that are not included in this publication.

The faster decline in mortality rates in younger age groups indicates a potential further reduction in mortality. The literature on the analysis of the mortality due to COPD in young age groups is limited. An analysis of the time trends in COPD mortality in the USA, published in 2015, pointed to a similar pattern of faster reduction in mortality among younger people [23].

The results of the above analysis may suggest that the implementation of successive health prevention decisions in Poland, i.e., smoke-free policies based on the WHO’s “MPOWER” strategy, which, between 2009 and 2017, contributed to a 9.4% decrease in the proportion of male smokers and a 16.7% decrease in female smokers as well as a significant reduction in passive exposure to tobacco smoke in public places [9], thereby ensured the downward trend of premature mortality due to COPD. Moreover, the beneficial changes were best seen in the younger age groups, who were eager to change their habits under the influence of health-promoting solutions. However, this work is designed as a descriptive epidemiological study. It aims to describe the trends in premature mortality due to COPD in Poland. The methodology used does not make it possible to show a cause-and-effect correlation between these factors. For this reason, the juxtaposition of COPD mortality with the prevalence of smoking is based on the results of previous analytical studies. The other limitation of this study is the potential underestimation of the number of deaths resulting from COPD in Poland. The disease is usually accompanied by other co-morbidities, e.g., cardiovascular diseases or cancers, which significantly affect the death statistics. For example, it is estimated that cardiovascular diseases are responsible for 25% of deaths of patients with diagnosed COPD [24]. A further potential cause of why COPD-related mortality is underestimated is the average quality of data on the cause of death as specified in death certificates. Apparently, the cause of every fourth death in Poland is registered by a “garbage code”, which makes it impossible to accurately identify the disease responsible for the death [25].

## 5. Conclusions

The study showed a steady downward trend in premature mortality due to chronic obstructive pulmonary disease in Poland in both sexes. The reduction in mortality was high, even though mortality due to this condition was still lower than in other European countries.

An effective long-term anti-smoking policy has resulted not only in a significant reduction in premature mortality due to COPD. It has also allowed Poland to effectively implement the objective of reducing premature mortality related to non-infectious diseases by 2025 as recommended by the WHO.

## Figures and Tables

**Figure 1 medicina-57-00126-f001:**
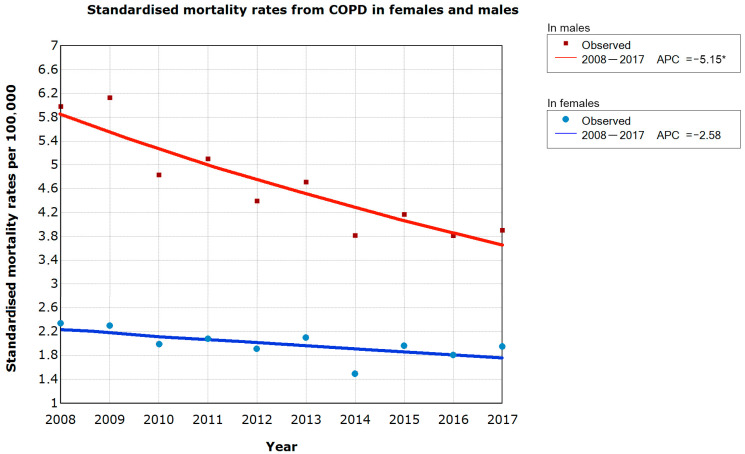
Time trends in standardised mortality rates from COPD in females and males. * the significant values were marked at *p* < 0.05.

**Table 1 medicina-57-00126-t001:** All and premature deaths caused by COPD by gender (absolute numbers and standardised mortality rates of all deaths and premature deaths).

GENDER	YEAR									
**WOMEN**	**2008**	**2009**	**2010**	**2011**	**2012**	**2013**	**2014**	**2015**	**2016**	**2017**
NUMBER OF ALL DEATHS	2365	2475	2095	2191	2202	2369	2002	2426	2227	2489
NUMBER OF PREMATURE DEATHS	373	382	346	367	345	383	272	385	335	351
STANDARDISED MORTALITY RATES	13.7	14.18	11.83	12.19	12.09	12.81	10.72	12.8	11.44	12.62
STANDARDISED MORTALITY RATES (0–64)	2.34	2.3	1.99	2.08	1.91	2.1	1.5	1.96	1.81	1.94
	**YEAR**									
**MEN**	**2008**	**2009**	**2010**	**2011**	**2012**	**2013**	**2014**	**2015**	**2016**	**2017**
NUMBER OF ALL DEATHS	5406	5549	4628	4946	4574	4813	3999	4301	3875	4278
NUMBER OF PREMATURE DEATHS	820	892	754	813	710	772	628	687	628	642
STANDARDISED MORTALITY RATES	52.2	52.58	43.13	45.21	41.55	42.95	35.07	37.11	31.98	34.92
STANDARDISED MORTALITY RATES(0–64)	5.98	6.13	4.83	5.1	4.39	4.71	3.81	4.17	3.81	3.9

**Table 2 medicina-57-00126-t002:** Premature mortality of the Polish population due to COPD by gender and age in 2008 (crude rates/100,000 population).

GENDER	AGE								
	20–24	25–29	30–34	35–39	40–44	45–49	50–54	55–59	60–64
WOMEN	0.07	0.06	0	0.16	0.6	1.35	3.99	9.09	14.29
MEN	0.13	0	0.27	0.47	1.27	3.44	8.1	17.5	45.4

**Table 3 medicina-57-00126-t003:** Mortality trends due to COPD among the Polish population by gender and age.

AGE	YEARS	WOMEN		YEARS	MEN	
		APC (95% CI)	AAPC (95% CI)		APC (95% CI)	AAPC (95% CI)
20–39	2008–2017	0	0	2008–2017	0	0
40–44	2008–2017		−5.7 (−19.8, +10.7)	2008–2017	−8.9 (−19.6, +4)	−8.9 (−19.6, +4)
	2008–2010	−44.8 (−76.7, +30.6)				
	2010–2017	+9.8 (−2.1, +23.2)				
45–49	2008–2017	−5.3 (−13.6, +3.9)	−5.3 (−13.6, +3.9)	2008–2017	−7.8 * (−10.2, −5.2)	−7.8 * (−10.2, −5.2)
50–54	2008–2017	−9.5 * (−13.5, −5.2)	−9.5 * (−13.5, −5.2)	2008–2017	−7.4 * (−9.5, −5.3)	−7.4 * (−9.5, −5.3)
55–59	2008–2017	−5.5 * (−7.9, −3.1)	−5.5 * (−7.9, −3.1)	2008–2017	−4.6 * (−6.9, −2.2)	−4.6 * (−6.9, −2.2)
60–64	2008–2017	+0.6 (−2.9, +4.4)	+0.6 (−2.9, +4.4)	2008–2017	−4.6 * (−6.8, −2.3)	−4.6 * (−6.8, −2.3)

The relevant rows of Table 3 contain detailed information on annual percentage change (APC) and average annual percentage change (AAPC) in mortality of women and man in the analysed period according to the analysed five-year age ranges. * the significant values were marked at *p*< 0.05.

## Data Availability

Data available in a publicly accessible repository that does not issue DOIs Publicly available datasets were analyzed in this study. This data can be found here: http://demografia.stat.gov.pl/bazademografia/Tables.aspx.

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
