# Peer review of "Premature Mortality Due to Chronic Obstructive Pulmonary Disease (COPD) in Poland"

_medicina, 2021, doi:10.3390/medicina57020126_

Round 1

Reviewer 1 Report

My prior comments have been addressed

Author Response

Dear Sir/Madam

Thank you very much for your review.

I consider your remarks very valuable. I believe that thanks to the advice provided, the work becomes clearer and more valuable for the readers.

Reviewer 2 Report

The authors have nicely addressed concerns for the table and figure presentation. However, the methodology and discussion still require modifications. The discussion includes a limitations paragraph as requested but the opening 2 paragraphs are more introduction and not discussion. Equally, the methodology should be expanded

Author Response

Dear Sir/Madam

Thank you for your comment.

Below please find my detailed response to your remarks.

  1. Methodology has been expended in lines 70-80.
  2. Discussion has been expended in the lines 144-167

This manuscript is a resubmission of an earlier submission. The following is a list of the peer review reports and author responses from that submission.

Round 1

Reviewer 1 Report

The authors look at COPD deaths in Poland among decedents < 65 - showing a decreasing trend from 2008-2017.

Major concerns:

It is not clear  what is reported is due to decreasing rates of COPD or people living longer ( or better therapies).  While this might be impossible to fully disentangle- including ALL decedents in the analysis would help a bit  ( and one could contrast the mortality rates in different age groups)

The overall mortality rates seemed a bit low-  how does this compare to other countries ( or other causes of death in Poland) ( in the US it is 30/100,000 - but that includes all age groups)

In the figures- both should use use the same scale.

Table 1 was not interpretable for me- this need to be redone to be understandable.

Author Response

Dear Sir/Madam

Thank you very much for your review and valuable comments. I have modified the content of the manuscript to implement your suggestions.

Please find my detailed replies to the individual passages of the article you have indicated. The changes have also been made in the paper.

1.

It is not clear  what is reported is due to decreasing rates of COPD or people living longer ( or better therapies).  While this might be impossible to fully disentangle- including ALL decedents in the analysis would help a bit  ( and one could contrast the mortality rates in different age groups)

In this paper we describe the change in premature mortality due to COPD. When using the proposed method, it is impossible to determine the direct cause of the changed mortality, thus we have described this limitation in the discussion section of the paper (170-173). Acting on your advice, we have included the data on COPD mortality in total population.

2.

The overall mortality rates seemed a bit low-  how does this compare to other countries ( or other causes of death in Poland) ( in the US it is 30/100,000 - but that includes all age groups)

Mortality due to COPD in Poland is at a fairly low level. According to the OECD report, HEALTH AT A GLANCE: EUROPE 2018, the average standardised mortality rate due to COPD for 28 European Union countries in 2015 amounted to 36.3/100 000. At the same time, the rate for the Polish populationwas 23.2/100 000.

This is quite astonishing, given that mortality rates due to most tobacco-dependent diseases are higher than the averages for countries in the European community.

3.

In the figures- both should use the same scale.

Figures1 and 2 have been redesigned to provide better clarity. They have been combined into a single, large figure.

4.

Table 1 was not interpretable for me- this need to be redone to be understandable.

Two additional tables have been added as follows:

Table 1. Absolute numbers and standardized mortality rates of all deaths and premature deaths.

Table 2. Exact mortality rates within the Polish population due to COPD by gender and age in 2008 as a benchmark for further analysis.

Table 3. (previously Table 1.) has been redesigned. Now it presents information on annual percentage change (APC) and average annual percentage change (AAPC) in male and female mortality in five year age ranges.

Reviewer 2 Report

Moryson and Stawinska-Witoszynska investigated the change in premature mortality rates of men and women due to COPD in Poland between 2008-2017, as smoking rates are reducing within Poland. The authors observe a significant drop in male deaths under the age of 65 and a trend reduction in female deaths for the same period. These changes were age-dependent. Overall, this is an interesting topic for the readers in a relevant subject matter. However, there are several minor issues that should be addressed. These items are listed below:

  1. It would be helpful if the scale in the y-axis for both figure 1 and 2 were the same. Equally, both graphs could be presented in one figure rather than 2.
  2. Please be mindful of font text size in graphs. It is very difficult to read the text in the figures
  3. What does the “Year of Change” column in Table 1 signify? Please clarify or remove
  4. The methodology is limited. Please give greater detail, specific details on how the data in table 1 were analyzed. Please also give the reader a little more information about the total population numbers and the total numbers of deaths per year for this analysis. These numbers are mentioned in the results text but could be incorporated into the table. Also, in table 1 it would be better to group 20-39 into one row as it is zero for <39.
  5. The limitation section on lines 75-82 should be in the discussion and not the methodology
  6. Overall, the discussion needs major changes. The main outcomes of this paper are not summarized and discussed in depth. Equally, more limitations are needed to be outlined in this section. Please redraft the discussion.

Author Response

Dear Sir/Madam

Thank you very much for your review.

I consider your remarks very valuable. I haveanalysed them with great diligence and modified the content of the manuscript to implement them. I believe that thanks to the advice provided, the work becomes clearer and more valuable to the readers.

Please find my detailed replies to the individual passages of the article you have indicated. The changes have also been introduced in the paper.

1

It would be helpful if the scale in the y-axis for both figure 1 and 2 were the same. Equally, both graphs could be presented in one figure rather than 2.

Figures1 and 2 have been redesigned to provide better clarity. They have been combined into a single, large figure.

2

Please be mindful of font text size in graphs. It is very difficult to read the text in the figures

The graphs have been redesigned to provide better legibility.

3

What does the “Year of Change” column in Table 1 signify? Please clarify or remove

The column in question (Year of Change) signifies the year in which, according to Joinpoint regression, a significant change of mortality time trend was observed. However, I do admit that this additional information may distort the clarity of the presented data, especially considering that the analysed time intervals are described in the afore mentioned table. Thus, the column in question has been removed from the table.

4

The methodology is limited. Please give greater detail, specific details on how the data in table 1 were analyzed. Please also give the reader a little more information about the total population numbers and the total numbers of deaths per year for this analysis. These numbers are mentioned in the results text but could be incorporated into the table. Also, in table 1 it would be better to group 20-39 into one row as it is zero for <39.

The specific details on the analysis of the data presented in table 3 (previously table 1) have been added.

We have also added information on the absolute numbers and standardized mortality rates of all deaths and premature deaths and we have placed them in newly created Table 1. Moreover, we have added Table 2 with exact mortality rateswithin the Polish population due to COPD by gender and age in 2008 as a benchmark for further analysis.

The groups 20-39 has been merged into one row of table 3 (previously Table 1).

5

The limitation section on lines 75-82 should be in the discussion and not the methodology

The limitations section has been included in the discussion (lines 170-181)

6

Overall, the discussion needs major changes. The main outcomes of this paper are not summarized and discussed in depth. Equally, more limitations are needed to be outlined in this section. Please redraft the discussion.

The discussion section has been further expanded. The main results of the paper have been discussed in more detail. The part of the discussion dealing with the limitations of the work has also been expanded.